# Trends and Regional Differences in the Prevalence of Dyslipidemia before and after the Great East Japan Earthquake: A Population-Based 10-Year Study Using the National Database in Japan

**DOI:** 10.3390/ijerph20010560

**Published:** 2022-12-29

**Authors:** Shigetaka Nakajima, Eri Eguchi, Narumi Funakubo, Fumikazu Hayashi, Masumi Iwai-Takano, Tetsuya Ohira

**Affiliations:** 1Department of Epidemiology, Fukushima Medical University School of Medicine, 1-1 Hikariga-oka, Fukushima 960-1295, Japan; 2Radiation Medical Science Center for the Fukushima Health Management Survey, Fukushima Medical University, 1 Hikariga-oka, Fukushima 960-1295, Japan

**Keywords:** earthquake, evacuation, dyslipidemia, man-made disasters, physical activity

## Abstract

Since the Great East Japan Earthquake in March 2011, an increase in lifestyle-related diseases due to changes in living environment following the nuclear power plant accident has been reported in Fukushima Prefecture, especially among evacuees. However, no long-term studies covering the entire Fukushima Prefecture have been conducted. The study aim was to investigate the effects of post-disaster evacuation life on the prevalence of dyslipidemia in Fukushima Prefecture using a national database. The data from 3,866,674 people who underwent specific health checkups between fiscal year (FY) 2008 and FY2017 were analyzed. Fukushima Prefecture was divided into four areas, and the prevalence of dyslipidemia and related parameters were compared. The prevalence of dyslipidemia increased overall, with a particularly sharp increase after FY2011 in the evacuation area. The sex- and age-adjusted odds ratio (95% confidence intervals) of having dyslipidemia in the evacuation area compared with that in the control area was 0.951 (0.929–0.973) in FY2008–2010, which increased to 1.130 (1.105–1.155) in FY2012–2014 and 1.117 (1.092–1.143) in FY2015–2017. Since the prevalence of dyslipidemia has increased and remained high after the earthquake in Fukushima Prefecture, especially in the evacuation area, continued measures to prevent cardiovascular diseases among the residents are needed.

## 1. Introduction

The Great East Japan Earthquake (GEJE) that occurred on 11 March 2011 caused widespread damage in the Tohoku region, including collapsed houses and disrupted lifelines, as well as tsunami damage along the Pacific coast, resulting in the evacuation of many residents [1]. In Fukushima Prefecture, in addition to the earthquake and tsunami, the accident at the Fukushima Daiichi Nuclear Power Plant (NPP) caused the evacuation of up to 100,000 people within the prefecture and 60,000 people outside the prefecture, for a total of approximately 160,000 people, and more than 30,000 people are still living as evacuees [2,3,4].

Following the disaster, the government designated the area within a 20–30-km radius of the NPP as the evacuation area and the area near the 30-km radius, where high-dose exposure (≥20 mSv/y) is expected to occur, as the planned evacuation area. Approximately 12% of the Fukushima Prefecture was designated as the evacuation area. The Government of Japan, Fukushima Prefecture, and local municipalities provided intangible and tangible support to evacuation areas. On the soft side, prefectural health surveys were conducted to ascertain the health status of residents, and health support activities included health classes and counseling [5], and visits to public housing facilities [6]. Partial exemption of copayment for medical expenses has also been provided [7]. Regarding hardware support, support was provided to reopen medical institutions and for the construction of new public medical institutions [8]. Comparing the number of medical institutions in evacuation areas before the earthquake and in January 2022, hospitals accounted for 25% (2 of 8), clinics for 45% (27 of 60), and dental clinics for 28.1% (9 of 32) [9]. Despite the various lines of support, evacuation has introduced health hazards to the residents, and the prevalence of obesity [10], hypertension [11], diabetes [12], dyslipidemia [13], liver dysfunction [14], atrial fibrillation [15], and other disorders has increased [16].

Compared with the national average of cardiac disease as a cause of death, Fukushima Prefecture has a significantly higher proportion of these deaths, and it is important to take measures against arteriosclerotic diseases that cause heart disease and lifestyle-related diseases that trigger heart disease [17]. Most of the surveys and studies on the deteriorating health status of evacuees reported to date have had a short observation period after the earthquake, so only a few studies have examined long-term trends [16]. Additionally, because the earthquake effects on the deterioration of the mental and physical health of residents in the evacuation area have not been compared with those in other areas, it is unknown if the deterioration of health after the earthquake was unique to the evacuation area residents.

The second phase of Health Japan 21 policy, which was launched in 2000 with the aim to promote health [18], commenced in 2013. The second interim report published in 2018 showed improvement in the prevalence of hypertension and diabetes; however, no improvement was observed for dyslipidemia [19]. Another short-term study reported an increase in dyslipidemia prevalence among residents of evacuation areas after the earthquake [13], albeit only for a short period of time. Given that dyslipidemia is one of the most important risk factors for cardiovascular disease, long-term follow-up of dyslipidemia prevalence is necessary to prevent cardiac disease in the evacuated areas.

Therefore, the aim of this study was to investigate the effects of post-disaster evacuation life on the prevalence of dyslipidemia in Fukushima Prefecture using a national database. We investigated the data on residents in the evacuation area and in all of Fukushima Prefecture, with an observation period of 10 years from fiscal year (FY) 2008 to FY2017, which included the pre-earthquake period. By setting a long observation period, we sought to not only understand the pre-disaster conditions, but also to examine regional differences in long-term changes in the effects of the disaster. In addition to the long-term observation period, the entire Fukushima Prefecture was included in this study, with the goal of identifying the influence of the radiation accident on both the entire Fukushima Prefecture and the specific evacuation area.

## 2. Materials and Methods

### 2.1. Study Population

We used the National Database of Health Insurance Claims and Specific Health Checkups of Japan (NDB)—a national claims database managed by the Japanese Ministry of Health, Labor, and Welfare (MHLW) [20]—to obtain data on health checkups in the Fukushima Prefecture from FY2008 to FY2017. In Japan, specific health checkups, which comprise two pillars of health examination and health guidance, have been implemented since FY2008 with a focus on metabolic syndrome to prevent lifestyle-related diseases. The subjects of the specific health checkups include insured individuals aged 40–74 years, including those who will reach the age of 40 years within the current fiscal year, and their dependents. The national examination rate was 38.9% in FY 2008 and 48.6% in FY 2014 [20], and the number of examinees has been increasing by approximately 1 million per year. The MHLW has collected anonymized data of specific health checkups and electronic health insurance claims; the data were publicly released as the NBD in 2011 for use by researchers. Only researchers in national or local government agencies, universities, and other quasi-public institutions are eligible to apply for the use of NDB, and a research plan must be approved by the MHLW Advisory Committee. Data extracted from the NDB are provided for research use after anonymization of information on specific patients and medical facilities; the NDB contains vast amounts of information, including patient age, sex, diagnosis, inpatient and outpatient medical data, and dental services, and the present study utilized data on specific health checkups. In the current study, we used data from a total of 3,866,674 persons in the basic resident registry of Fukushima Prefecture and who received the specific health checkups during the 10-year period from FY2008 to FY2017 for analyses in the present study.

During analysis of the data, we identified 6946 subjects who were missing at least one of the three items required for the determination of lipid abnormalities (triglyceride levels, high-density lipoprotein (HDL) levels, and medication for dyslipidemia) and one subject whose age category was not covered by the specific health checkups, and the subjects were were deleted. For the 50 subjects who had two records in the same year, only one record was used, and duplicate data were deleted, finally resulting in 3,859,677 subjects for analysis (Figure 1).

### 2.2. Ethical Considerations

The data used in this study were approved by the General Ethics Committee of Fukushima Medical University (General S30225). The study included the following:Prevalence of dyslipidemia and its trend over time.Comparison of trends over time between the evacuation area and other areas.Impact of evacuation on the prevalence of dyslipidemia.

### 2.3. Measurements and Definitions

The results of physical measurements, physical examination, blood-pressure measurement, blood test, urinalysis, and a 22-item self-administered questionnaire related to lifestyle of the specific health checkups were used [21]. From the blood test items, we used the results of triglyceride and HDL cholesterol measurements related to dyslipidemia, and to determine obesity, we used the results of BMI, calculated as weight (kg)/height (m)^2^, and abdominal circumference.

The questionnaire revealed that the respondents were taking cholesterol-lowering medication (0: none, 1: yes), eating faster than others (1: fast, 2: normal, 3: slow), eating before bedtime, eating dinner ≤2 h ≥3 days a week (0: no, 1: yes), eating a snack after dinner ≥3 days a week (0: no, 1: yes), skipping breakfast ≥3 times a week (0: no, 1: yes), exercising lightly ≥30 min at a time ≥2 days a week and ≥1 year (0: no, 1: yes), walking or equivalent physical activity in daily life ≥2 days a week for ≥1 year (0: no, 1: yes), walking faster than their peers of about the same age (0: no, 1: yes), currently smoke cigarettes habitually (0: no, 1: yes), drink alcohol more often (1: <180 mL, 2: 180–360 mL, 3: 360–540 mL, 4: ≥540 mL), and well rested with sleep (0: no, 1: yes).

Dyslipidemia was defined as having a triglyceride level ≥150 mg/dL, an HDL level <40 mg/dL, or taking cholesterol-lowering medication. Obesity was defined as a BMI ≥25 kg/m^2^ or an abdominal circumference ≥85 cm for men and ≥90 cm for women in accordance with the diagnostic criteria for metabolic syndrome in Japan [22].

To examine the differences in the impact of the evacuation on the health status of residents between the evacuation area and other areas in Fukushima Prefecture, we set up four areas according to whether or not people were evacuated after the disaster. Twelve municipalities were designated as comprising the evacuation area: Tamura City, Minamisoma City, Kawamata Town, Hirono Town, Naraha Town, Tomioka Town, Kawauchi Village, Okuma Town, Futaba Town, Namie Town, Katsurao Village, and Iitate Village, where most residents were evacuated under government orders after the NPP disaster. Of the seven secondary medical care areas in Fukushima Prefecture, the Aizu and Minamiaizu regions were designated as Aizu (mountain area), where most of the residents have not been evacuated. The remaining two areas where some residents voluntarily evacuated were designated as “Nakadori” (north, central, and south of Fukushima Prefecture), and Iwaki City, Soma City, and Shinchi Town as “Hamadori” (costal area) (Figure 2). Kawamata Town and Tamura City, which were included in the evacuation area, were excluded from Nakadori. Based on the premise that there was no overlap in the registered areas in the Basic Resident Ledger, the four areas of the evacuation area, Aizu, Nakadori, and Hamadori were treated as independent groups.

Four time phases were designated to examine the effects of post-disaster time on the association of evacuation and lifestyle with dyslipidemia. The period from FY2008 to FY2010, which was before the earthquake and unrelated to the evacuation, was defined as Phase 1. Phase 2 was defined as the period in FY2011, when people lived in evacuation shelters while receiving support, such as meals. The first 3 years from FY2012 to FY2014 were designated as Phase 3, and the second 3 years from FY2015 to FY2017 were designated as Phase 4, taking into account changes over time in the effects of the disaster (Table 1). In the present study, data for up to 3 years within each time phase were recorded for each participant. The total number of observations was reduced in the analysis because data from only one time point within a single time phase were used for each participant. For participants with multiple datapoints available for each time phase, data furthest away from the year of the earthquake were used in the analysis.

Fukushima Prefecture is divided into four regions, based on the consideration of secondary medical care areas and on whether evacuation orders have been issued.

### 2.4. Statistical Analysis

Age was divided into seven age groups (40–44, 45–49, 50–54, 55–59, 60–64, 65–69, and 70–74) for analysis. The mean triglyceride levels, HDL cholesterol levels, cholesterol-lowering medication rates, and prevalence of dyslipidemia were presented for each area for each year to identify trends in each area as well as trends specific to the evacuation area. To examine the percentage change in the prevalence of dyslipidemia in each area, we performed a joinpoint analysis using the Joinpoint Regression Program 4.9.0.0–March, 2021 (Statistical Research and Applications Branch, National Cancer Institute, USA) using age-adjusted data for men and women [23]. Regional differences in the mean triglyceride levels, mean HDL cholesterol levels, proportion of cholesterol-lowering medication use, and the prevalence of dyslipidemia were compared by time phase and sex.

For analyses, the Kruskal–Wallis test was used to compare triglyceride and HDL levels among the four regions, and significant differences among the four regions were assessed using Bonferroni’s test. Pearson’s χ-square test was used for comparisons of medication use and dyslipidemia prevalence among the four regions. In addition, multivariable-adjusted logistic regression analysis was performed to examine the associations of evacuation and lifestyle-related factors with the prevalence of dyslipidemia. Stata/SE 16 (Stata Corp4906 Lakeway Dr College Station, TX 77,846 USA) was used for analysis, and *p*-values of <0.01 were accepted as indicating statistical significance.

## 3. Results

### 3.1. Trend Analysis

Annual changes in triglyceride levels by area are shown in Figure 3 for males and females, respectively. For residents outside of the evacuation area, there was no significant change after FY2011 for males and females. For males in the evacuation area, the mean level of triglycerides increased once in FY2011 but decreased in FY2012, and then no longer differed from the other areas and remained at the same level as those in the other areas thereafter. For females in the evacuation area, there was a significant increase in FY2011, followed by a higher mean value than those in the other areas. Figure 4 shows the annual changes in HDL cholesterol levels for males and female, respectively. For males, there was an HDL cholesterol level increase in FY2011, a decrease in FY2012, and a gradual increase from FY2013 onward in all areas. The mean levels of HDL cholesterol for females in the evacuation area decreased in FY2012 but increased moderately thereafter, as in other areas.

As shown in Figure 5, the age-adjusted proportion of cholesterol-lowering medication use in the evacuation area increased rapidly from FY2011 to FY2013 for males and females, and it has since remained higher than in other areas for females. As shown in Figure 6, the age-adjusted proportion of the prevalence of dyslipidemia increased from FY2011 to FY2013 and has remained high from FY2014 onward. The evacuation area was characterized by a more pronounced increase from FY2011 to FY2013 than those in other areas, and the large difference in the prevalence of dyslipidemia that occurred at that time has remained, although it has narrowed slightly with each passing year. To identify changes in the prevalence of dyslipidemia, a joinpoint analysis was performed using the prevalence by year of examination adjusted for sex and age using the standard population. The year in which a significant change in the rate of increase or decrease in the prevalence of dyslipidemia over 1 year (hereafter referred to as “inflection point”) was observed in FY2012 in Aizu and in FY2013 in the other areas. The average annual rates of change before and after the inflection point were 2.16% and −0.16% in Aizu, 1.25% and 0.0% in Nakadori, 1.35% and −0.32% in Hamadori, and 4.16% and −0.16% in the evacuation area, respectively.

### 3.2. Associations of Evacuation and Lifestyle-Related Factors with Dyslipidemia

The χ-square test was used to confirm the differences in the prevalence of dyslipidemia between areas, and significant differences were found in all time phases (Table 2). For males, although the prevalence of dyslipidemia was highest in Hamadori (38.6%) and lowest in the evacuation area (36.2%) before the disaster, the prevalence was highest in the evacuation area in FY2012 (42.1%) and thereafter. For females, the prevalence of dyslipidemia was highest in Hamadori (24.6%) before the disaster and highest in the evacuation area after the disaster and thereafter (27.7% in FY2011, 32.3% in FY2012–2014, and 33.6% in FY2015–2017). For each of the four phases, the prevalence of dyslipidemia was summarized by region and a χ-square test was performed.

Table 3 shows changes in lifestyle in each region during each phase based on data obtained from the questionnaire. In the evacuation area, exercise habits and physical activity decreased in FY2011, followed with an increase since then, whereas the proportion of obese and fast eaters increased in FY2011 and continued to increase thereafter in the same area. In the overall Fukushima Prefecture, the proportion of people who skip breakfast has been increasing at a constant rate since before the GEJE. Annual trend of age-adjusted smoking rates in each region did not differ significantly between regions but did show different trends between males and females. The age-adjusted smoking rates have remained unchanged among males (FY2012; 36.4%, FY2013; 36.7%, FY2014; 37.1%, FY2015; 37.5%, FY2016; 37.4%, FY2017; 37.3%) and increased among females (FY2012; 9.5%, FY2013; 10%, FY2014; 10.3%, FY2015; 10.7%, FY2016; 10.8%, FY2017; 11.2%) since FY2011.

In the age- and sex-adjusted logistic regression analysis, the prevalence of dyslipidemia was significantly lower in Nakadori than in Aizu and significantly higher in Hamadori both before and after the disaster. However, the prevalence was significantly lower in the evacuation area before the disaster than in Aizu but significantly higher in FY2012–2014 and FY2015–2017 after the disaster, and the adjusted Odds Ratios (ORs) (95% confidence intervals (CIs)) were 0.951 (0.929–0.973) in FY2008–2010, 1.130 (1.105–1.155) in FY2012–2014, and 1.117 (1.092–1.143) in FY2015–2017. Obesity and lifestyle habits, such as eating fast, skipping breakfast, smoking, and heavy alcohol consumption, were associated with an increased prevalence of dyslipidemia in all time phases, whereas eating slowly, regular exercise, and increased physical activity were associated with a lower prevalence of dyslipidemia (Table 4). In the multivariable-adjusted analysis, these associations were essentially the same (Table 5). The adjusted ORs (95%Cis) of the prevalence of dyslipidemia for residents in the evacuation area compared with those in Aizu were 0.915 (0.943–0.981) in FY2008–2010, 1.074 (1.045–1.105) in FY2012–2014, and 1.053 (1.023–1.083) in FY2015–2017, respectively.

## 4. Discussion

Although there have been several reports of a significant increase in the prevalence of dyslipidemia among residents of the evacuation area after the NPP accident, there have been no comparisons between the evacuation area and other areas [13,24,25]. In this study, we examined the prevalence of dyslipidemia before and after the disaster throughout Fukushima Prefecture, including areas outside of the evacuation area, using the NDB and found that although a gradual increase in the prevalence of dyslipidemia was observed after the disaster throughout Fukushima Prefecture, a rapid increase in the prevalence was observed from FY2011 to FY2013 in the evacuation area, and the prevalence has continued to be significantly higher than in areas not affected by the disaster. Therefore, it can be assumed that the risk of cardiovascular diseases remains high among evacuation area residents.

The Increase in the prevalence of dyslipidemia from FY2011 to FY2013 was observed in other areas as well, although the proportion was smaller than in the evacuation area, suggesting that factors other than evacuation may have had an impact. In addition to genetic predisposition and sex, lifestyle factors, such as stress, eating habits, exercise habits, smoking, and alcohol consumption are known to have a significant role in the development of dyslipidemia [26]. In Fukushima Prefecture, many residents tended to refrain from their outdoor activities in the months following the accident. This may have affected the amount of physical activity and incidence of dyslipidemia.

The results of this study showed that the prevalence of dyslipidemia increased significantly after the disaster, especially in the evacuation area, for the following three possible reasons. The first reason is the effect of the post-disaster decline in exercise habits and physical activity in this study, and previous epidemiological studies conducted in the evacuation area of Fukushima Prefecture showed that high physical activity was associated with a lower risk of developing dyslipidemia, and increasing body weight was associated with a higher risk of developing dyslipidemia after the disaster [13,25].

Second, dietary changes among evacuees may have influenced the increase in dyslipidemia. In the evacuation area in Miyagi Prefecture, many residents lived in shelters immediately after the disaster, and the meals provided in shelters were reported to be low in vitamins and high in carbohydrates and fats [27], and in Fukushima Prefecture, living in non-home conditions was associated with poor dietary intake of fruits and vegetables (non-juice), meat, soybean products, and dairy products [28]. Furthermore, a previous study conducted among evacuees of Fukushima Prefecture reported that a dietary pattern with a high intake of vegetables and soy products was associated with a lower risk of prevalence of dyslipidemia and obesity [29]. Therefore, it may have influenced the increase in the prevalence of dyslipidemia in the evacuation area. Meanwhile, although eating speed and skipping breakfast were associated with the prevalence of dyslipidemia, the distribution of eating speed changes only slightly from year to year, with small differences between areas, and the proportion of people who skip breakfast increased gradually each year. Therefore, these dietary behaviors may have influenced the increase in the prevalence of dyslipidemia in Fukushima Prefecture, but it is difficult to explain the difference in the prevalence between the evacuation area and other areas by these habits. Third, psychological factors may have affected the regional differences in the prevalence. Previous studies have shown that the proportion of people with high psychological distress assessed using the Kessler Psychological Distress Scale (K6) was higher in Fukushima Prefecture than in Miyagi Prefecture [30], suggesting that in addition to the earthquake and tsunami, the Fukushima Daiichi NPP accident may have had additional effects on the residents of Fukushima Prefecture. Stress due to man-made disasters reportedly has a significant impact [31], and a systematic review and meta-analysis for psychological stress and metabolic syndrome showed that perceived stress was associated with increased body weight, waist circumference, triglycerides, and decreased HDL cholesterol [32]. Therefore, from FY 2011 to FY 2013, the prevalence of dyslipidemia among the evacuation area residents may have increased dramatically because of health concerns caused by the NPP accident, and stress related to life concerns, such as unemployment, compensation, and guarantees in the evacuation area, also may have increased. Indeed, a survey among residents of the evacuation area in Fukushima Prefecture found that a significantly higher proportion of those with higher scores on PTSD-related symptoms had metabolic syndrome, with a particularly strong association with women [33]. In the present study, the association between evacuation and dyslipidemia was somewhat stronger for females than for males, supporting previous findings.

In this study, smoking and heavy alcohol consumption were both associated with the prevalence of dyslipidemia. Smoking rates have remained unchanged among males and increased among females since FY2011, and it is possible that the increased smoking rate among females increased the prevalence of dyslipidemia in Fukushima Prefecture as a whole. Small amounts of alcohol consumption increase HDL cholesterol levels, whereas heavy alcohol consumption increases triglyceride levels. Thus, the present study results suggests that small amount of alcohol consumption reduces the odds ratio for dyslipidemia by increasing HDL levels and that heavy alcohol consumption increases the odds ratio for dyslipidemia by increasing triglyceride levels. However, despite a trend toward increased frequency and amount of alcohol consumption, there were no regional differences in smoking and alcohol consumption within Fukushima prefecture, suggesting that the changes in smoking rate and alcohol consumption did not affect the regional differences in dyslipidemia. However, the increase in the prevalence of dyslipidemia in the evacuation area may be partially due to the increased proportion of people taking cholesterol-lowering medications. This may be due to the medical cost-sharing exemption system for residents in the evacuation area that started after the NPP disaster. However, the fact that triglyceride levels remained higher in Fukushima prefecture than in other areas despite the increase in the proportion of people taking such medications suggests that dyslipidemia remains not well controlled among the evacuation area residents relative to those in other areas.

However, the joinpoint analysis revealed that the year with significant 1-year changes in dyslipidemia prevalence was FY2012 for Aizu and FY2013 for the other regions. This finding indicates that dyslipidemia prevalence significantly increased during the first two years after the earthquake. In other words, in addition to the abovementioned lifestyle and psychological issues, social issues such as medical resources might have influenced the increase in dyslipidemia prevalence. After the earthquake, many medical facilities were damaged, hindering access to medical services and directly or indirectly affecting the quality of medical care that was provided. Therefore, in addition to the temporary difficulty in receiving medical care experienced by many evacuees, it is possible that disease control became worse as a result of medical institutions in Fukushima Prefecture not being adequately prepared to receive patients. Indeed, decreased medication adherence and poorer control of diabetes and hypertension have been reported after the earthquake [34]. Furthermore, emergence and spread of the coronavirus disease 2019 pandemic since 2020 have resulted in extended periods of restricted contact with the outside world, which might have resulted in an increase in the number of residents with reduced physical activity and restricted access to medical care; these effects should be examined in future studies.

The strength of this study is that the subjects included not only residents of the evacuation area but also all insured Fukushima Prefecture residents aged ≥40 years old, allowing the comparison of regional differences among a large resident-based population. There were also several limitations that should be considered. First, LDL cholesterol is not a criterion for metabolic syndrome in the specific health checkup of Japan and was not included as a parameter in the present study. However, further analysis in future studies is warranted since LDL cholesterol is an important risk factor for atherosclerosis. Second, the rate of the specific health checkups was only about 48.4% (FY2014), so we do not know if these results are completely representative of all residents. In addition, not all residents were examined every year during the 10-year observation period. Third, since the evacuation area was defined as the municipalities in the areas that had to be evacuated, not all residents were evacuated. Therefore, the difference in the prevalence between the evacuation area and other areas may have been underestimated. Finally, when comparing the evacuation area with other areas, only measurements from the specific health checkups and no others, such as diet, socioeconomic factors, and mental health status, were used as confounding factors for adjustment. Dietary factors in particular were strongly associated with dyslipidemia and should be examined in the future.

## 5. Conclusions

We examined the trends in dyslipidemia among residents of Fukushima Prefecture, which was severely affected by the NPP accident that followed the GEJE, over a 10-year period before and after the disaster, and found that the prevalence of dyslipidemia was significantly lower among residents of the evacuation area before the disaster than in the control area (Aizu), which was less affected by the disaster. The prevalence of dyslipidemia has been continuously high since FY2012 after the disaster. The long-term regional differences were influenced by the rapid increase in the proportion of people with dyslipidemia in the affected areas after the disaster, especially from FY2011 to FY2013. Lifestyle habits, such as obesity, eating fast, skipping breakfast, lack of exercise/physical inactivity, smoking, and heavy drinking, were associated with dyslipidemia; therefore, to reduce the prevalence of dyslipidemia among residents of Fukushima Prefecture, especially in the evacuation area, it is necessary to improve lifestyle habits, such as smoking cessation and alcohol consumption, to eliminate obesity, and to consider how to improve physical activity, which has been declining.

## Figures and Tables

**Figure 1 ijerph-20-00560-f001:**
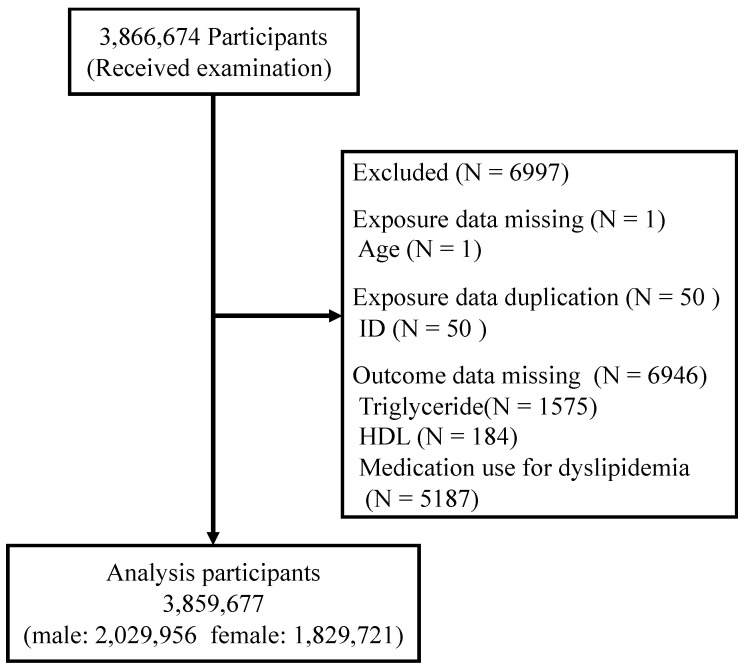
Flowchart of the target population. Total number of people included in the study analysis, with breakdown of the study cohort according to sex, within the total participants for whom data were obtained on specific health checkups for the ten-year period from fiscal year 2008 to fiscal year 2017.

**Figure 2 ijerph-20-00560-f002:**
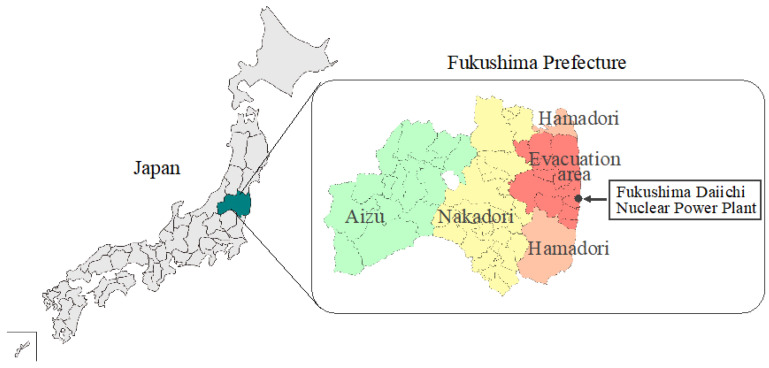
Regional Classification in Fukushima Prefecture.

**Figure 3 ijerph-20-00560-f003:**
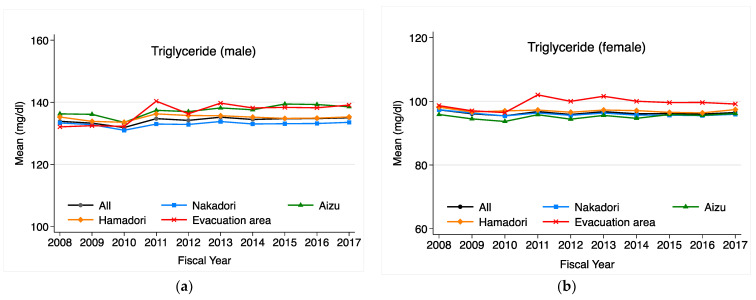
(**a**) Annual trends in age-adjusted mean triglyceride levels for males in each region. (**b**) Annual trends in age-adjusted mean triglyceride levels for females in each region. In addition to the four regions, trends for Fukushima Prefecture as a whole are also shown.

**Figure 4 ijerph-20-00560-f004:**
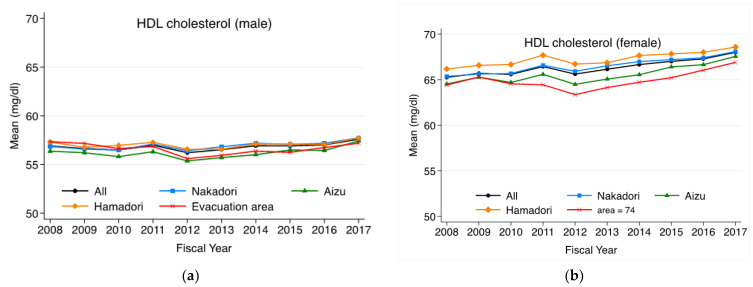
(**a**) Annual trends in age-adjusted mean HDL cholesterol levels for males in each region. (**b**) Annual trends in age-adjusted mean HDL cholesterol levels for females in each region. In addition to the four regions, trends for Fukushima Prefecture as a whole are also shown.

**Figure 5 ijerph-20-00560-f005:**
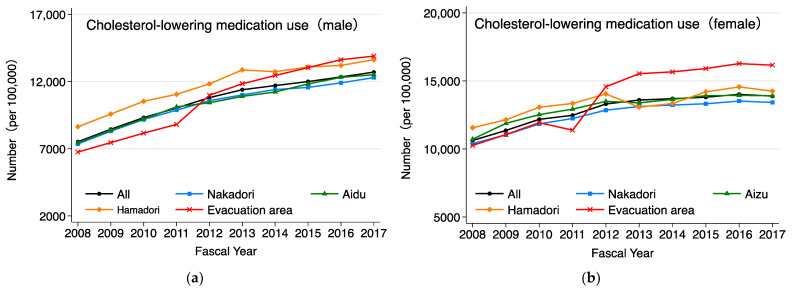
(**a**) Annual trend of age-adjusted rate of cholesterol-lowering medication use for males in each region. (**b**) Annual trend of age-adjusted rate of cholesterol-lowering medication use for females in each region. In addition to the four regions, trends for Fukushima Prefecture as a whole are also shown.

**Figure 6 ijerph-20-00560-f006:**
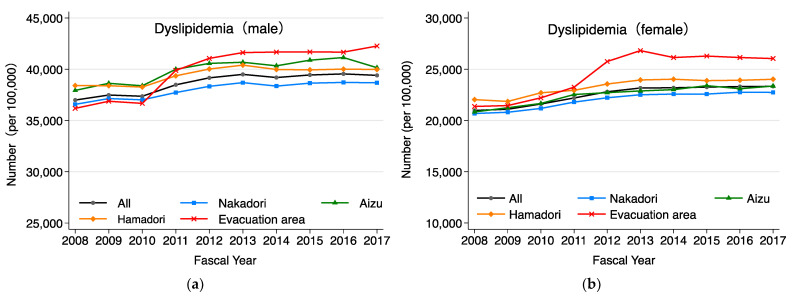
(**a**) Annual trend of age-adjusted dyslipidemia prevalence in males in each region. (**b**) Annual trend of age-adjusted dyslipidemia prevalence in females in each region. In addition to the four regions, trends for Fukushima Prefecture as a whole are also included.

**Table 1 ijerph-20-00560-t001:** Time phase setting.

Phase	Fiscal Year	Contents	Number of Participants
1	2008–2010	Period before the earthquake and unrelated to the evacuation	611,247
2	2011	Period during which people lived in evacuation shelters while receiving support such as meals	356,795
3	2012–2014	Period of post-disaster evacuation life outside the evacuation center (early)	668,996
4	2015–2017	Period of post-disaster evacuation life outside the evacuation center (late)	594,509

Time phases were established based on presence of evacuees, environment of evacuees, and time elapsed since evacuation.

**Table 2 ijerph-20-00560-t002:** Regional comparison of the prevalence of dyslipidemia in each phase.

Phase (Fiscal Year)	Sex	Nakadori (%)	Aizu (%)	Hamadori (%)	Evacuation Area (%)	χ^2^	*p* Value
1 (FY2008–FY2010)	male	37.0	38.2	38.6	36.1	80.4	<0.001
	female	22.9	24.2	24.6	24.3	85.8	<0.001
2 (FY2011)	male	37.8	39.9	39.5	39.8	70.9	<0.001
	female	25.2	27.3	26.7	27.7	87.9	<0.001
3 (FY2012–FY2014)	male	38.9	40.8	40.5	42.1	173.4	<0.001
	female	26.2	26.8	27.3	32.3	421.4	<0.001
4 (FY2015–FY2017)	male	39.4	41.1	40.7	42.7	142.4	<0.001
	female	27.1	28.8	28.5	33.6	461.0	<0.001

**Table 3 ijerph-20-00560-t003:** Regional differences in the proportion of lifestyle-related habits in each phase.

Lifestyle-Related Habits	Phase (Fiscal Year)	Aizu (%)	Nakadori (%)	Hamadori (%)	Evacuation Area (%)	All (%)
Exercise habit, yes	1 (FY2008–FY2010)	23.9	26.0	29.0	24.5	26.0
	2 (FY2011)	24.4	25.1	27.3	22.0	25.1
	3 (FY2012–FY2014)	23.1	25.4	28.7	26.4	25.6
	4 (FY2015–FY2017)	23.1	25.3	27.3	26.8	25.4
Physical activity, yes	1 (FY2008–FY2010)	32.2	34.0	34.8	31.8	33.6
	2 (FY2011)	33.3	33.6	34.8	30.1	33.4
	3 (FY2012–FY2014)	31.1	34.5	36.1	32.0	34.0
	4 (FY2015–FY2017)	31.2	34.6	35.6	31.0	34.0
Walk faster, yes	1 (FY2008–FY2010)	43.1	45.4	41.0	40.3	43.9
	2 (FY2011)	44.6	45.8	39.4	40.3	44.2
	3 (FY2012–FY2014)	43.7	45.8	38.6	40.7	44.0
	4 (FY2015–FY2017)	43.4	45.0	38.6	39.9	43.2
Obesity, yes	1 (FY2008–FY2010)	40.6	39.0	42.1	41.1	40.0
	2 (FY2011)	40.2	38.7	42.1	43.3	39.9
	3 (FY2012–FY2014)	40.4	39.1	43.0	44.9	40.5
	4 (FY2015–FY2017)	41.5	40.5	42.9	45.9	41.5
Eating speed, fast	1 (FY2008–FY2010)	28.0	27.9	30.5	25.9	28.2
	2 (FY2011)	28.7	28.6	30.3	27.2	28.8
	3 (FY2012–FY2014)	29.0	28.5	30.3	27.9	28.8
	4 (FY2015–FY2017)	28.8	28.4	29.2	28.2	28.6
Eating speed, slow	1 (FY2008–FY2010)	6.7	7.4	7.1	7.5	7.2
	2 (FY2011)	6.4	7.2	6.7	8.0	7.1
	3 (FY2012–FY2014)	6.4	7.1	7.0	7.6	7.0
	4 (FY2015–FY2017)	6.5	7.3	6.9	7.7	7.2
Skip breakfast, yes	1 (FY2008–FY2010)	11.4	12.3	13.5	9.7	12.1
	2 (FY2011)	11.0	12.0	12.8	10.3	11.8
	3 (FY2012–FY2014)	13.5	14.1	14.0	12.2	13.8
	4 (FY2015–FY2017)	13.9	14.9	15.2	12.6	14.6
Current smoking, yes	1 (FY2008–FY2010)	26.1	26.6	28.7	25.1	26.8
	2 (FY2011)	23.6	23.7	26.2	24.1	24.1
	3 (FY2012–FY2014)	25.9	25.4	27.4	25.1	25.8
	4 (FY2015–FY2017)	25.2	24.8	26.3	24.6	25.1
Excess drinking (≥360 mL sake/day)	1 (FY2008–FY2010)	15.5	14.4	16.7	14.4	14.9
	2 (FY2011)	14.5	13.9	16.6	14.5	14.5
	3 (FY2012–FY2014)	15.7	14.7	18.0	15.5	15.4
	4 (FY2015–FY2017)	15.3	14.8	17.4	15.0	15.3
Age-adjusted smoking rates (male)	FY2012	37.8	35.7	37.1	38.1	36.4
	FY2013	37.8	36.0	37.1	38.9	36.7
	FY2014	38.3	36.5	37.5	38.4	37.1
	FY2015	38.6	37.0	37.7	38.7	37.5
	FY2016	38.6	36.9	37.6	38.9	37.4
	FY2017	38.2	36.7	37.7	39.7	37.3
Age-adjusted smoking rates (male)	FY2012	10	9.1	10.9	9.1	9.5
	FY2013	10.6	9.5	11.1	9.6	10
	FY2014	10.9	9.8	11.6	9.9	10.3
	FY2015	11.5	10.2	11.8	10.2	10.7
	FY2016	11.6	10.4	11.7	10.5	10.8
	FY2017	11.8	10.7	12.5	11	11.2

Exercise habit: those who have been performing light sweaty exercise for ≥30 min at a time, ≥2 days a week, for ≥1 year. Physical activity: those who walked or engaged in walking or equivalent physical activity in daily life ≥2 days per week for ≥1 year. Walk faster: those who walked faster than others of approximately the same age. Obesity was calculated using body mass index and abdominal circumference. Eating speed: how fast a person eats relative to others. Skip breakfast: those who skipped breakfast ≥3 days a week. Current smoking: those who smoked ≥100 cigarettes or for ≥6 months and were smoking for the last month. Excess drinking: alcohol equivalent to 500 mL beer, 60 mL double whiskey, or two glasses of wine (240 mL) in sake-equivalent volume.

**Table 4 ijerph-20-00560-t004:** Logistic regression analysis to determine the effect of specific factors on dyslipidemia prevalence with adjustment for age and sex.

		Phase 1 (FY2008–FY2010)		Phase 2 (FY2011)		Phase 3 (FY2012–FY2014)		Phase 4 (FY2015–FY2017)	
		N = 611,247		N = 356,795		N = 668,996		N = 594,509	
Factor	Contents	Odds Ratio (95% CI)	*p* Value	Odds Ratio (95% CI)	*p* Value	Odds Ratio (95% CI)	*p* Value	Odds Ratio (95% CI)	*p* Value
Area of residence	Aizu	1		1		1		1	
	Nakadori	0.969 (0.954–0.985)	<0.001	0.942 (0.924–0.961)	<0.001	0.964 (0.950–0.978)	<0.001	0.955 (0.940–0.970)	<0.001
	Hmadori	1.049 (1.028–1.070)	<0.001	1.018 (0.993–1.044)	0.152	1.034 (1.016–1.053)	<0.001	1.022 (1.003–1.042)	0.023
	Evacuation area	0.951 (0.929–0.973)	<0.001	1.015 (0.984–1.047)	0.348	1.130 (1.105–1.155)	<0.001	1.117 (1.092–1.143)	<0.001
obesity	Yes	2.626 (2.596–2.657)	<0.001	2.628 (2.589–2.667)	<0.001	2.720 (2.691–2.750)	<0.001	2.730 (2.699–2.762)	<0.001
Eating speed	Fast	1.271 (1.253–1.289)	<0.001	1.255 (1.234–1.276)	<0.001	1.238 (1.223–1.253)	<0.001	1.240 (1.224–1.256)	<0.001
	Normal	1		1		1		1	
	Slow	0.953 (0.938–0.968)	<0.001	0.822 (0.797–0.848)	<0.001	0.821 (0.802–0.840)	<0.001	0.820 (0.801–0.840)	<0.001
Eating before bedtime	Yes	1.043 (1.029–1.058)	<0.001	1.021 (1.004–1.040)	0.018	1.008 (0.995–1.021)	0.222	1.010 (0.997–1.024)	0.145
Night meal	Yes	1.025 (1.007–1.043)	<0.001	0.997 (0.975–1.020)	0.792	1.012 (0.996–1.029)	0.143	1.005 (0.988–1.022)	0.589
Skip breakfast	Yes	1.140 (1.119–1.162)	<0.001	1.115 (1.089–1.142)	<0.001	1.095 (1.077–1.113)	<0.001	1.073 (1.055–1.091)	<0.001
Exercise habits	Yes	0.943 (0.929–0.956)	<0.001	0.928 (0.912–0.945)	<0.001	0.939 (0.927–0.952)	<0.001	0.929 (0.917–0.941)	<0.001
Physical activity	Yes	0.838 (0.827–0.849)	<0.001	0.835 (0.821–0.848)	<0.001	0.839 (0.829–0.849)	<0.001	0.848 (0.837–0.858)	<0.001
Walking speed	Yes	0.903 (0.892–0.914)	<0.001	0.904 (0.891–0.918)	<0.001	0.892 (0.882–0.902)	<0.001	0.887 (0.877–0.898)	<0.001
Smoking	Yes	1.322 (1.305–1.340)	<0.001	1.302 (1.279–1.325)	<0.001	1.274 (1.258–1.290)	<0.001	1.264 (1.247–1.281)	<0.001
Frequency of drinking	Daily	1		1		1		1	
	Occasionally	1.170 (1.154–1.194)	<0.001	1.147 (1.124–1.171)	<0.001	1.183 (1.165–1.201)	<0.001	1.146 (1.128–1.164)	<0.001
	Almost never	1.443 (1.420–1.466)	<0.001	1.420 (1.392–1.449)	<0.001	1.439 (1.418–1.460)	<0.001	1.396 (1.375–1.417)	<0.001
Amount of sake	<180 mL	1		1		1		1	
	180–360 mL	0.888 (0.873–0.903)	<0.001	0.894 (0.875–0.912)	<0.001	0.870 (0.857–0.884)	<0.001	0.882 (0.868–0.896)	<0.001
	360–540 mL	1.101 (1.077–1.126)	<0.001	1.067 (1.038–1.097)	<0.001	1.036 (1.016–1.057)	<0.001	1.066 (1.044–1.088)	<0.001
	>540 mL	1.543 (1.485–1.603)	<0.001	1.453 (1.384–1.525)	<0.001	1.435 (1.386–1.485)	<0.001	1.423 (1.373–1.475)	<0.001
Rested by sleep	Yes	1.032 (1.018–1.046)	<0.001	1.05 (1.033–1.067)	<0.001	1.029 (1.017–1.042)	<0.001	1.032 (1.020–1.045)	<0.001

CI: confidence interval.

**Table 5 ijerph-20-00560-t005:** Multivariable logistic regression analysis to determine the effect of region of residence on dyslipidemia prevalence in each phase after adjusting for specific factors.

		Phase 1 (FY2008–FY2010)		Phase 2 (FY2011)		Phase 3 (FY2012–FY2014)		Phase 4 (FY2015–FY2017)	
		N = 611,247		N = 356,795		N = 668,996		N = 594,509	
Factor	Contents	Odds Ratio (95% CI)	*p* Value	Odds Ratio (95% CI)	*p* Value	Odds Ratio (95% CI)	*p* Value	Odds Ratio (95% CI)	*p* Value
Area of residence	Aizu	1		1		1		1	
	Nakadori	1.002 (0.982–1.023)	0.847	0.944 (0.922–0.966)	<0.001	0.969 (0.952–0.986)	<0.001	0.957 (0.940–0.975)	<0.001
	Hamadori	0.984 (0.959–1.010)	0.222	0.918 (0.890–0.947)	<0.001	0.974 (0.952–0.998)	0.031	0.957 (0.935–0.979)	<0.001
	Evacuation area	0.915 (0.888–0.943)	<0.001	0.981 (0.943–1.019)	0.32	1.074 (1.045–1.105)	<0.001	1.053 (1.023–1.083)	<0.001
Obesity	Yes	2.608 (2.570–2.646)	<0.001	2.602 (2.556–2.649)	<0.001	2.632 (2.598–2.668)	<0.001	2.648 (2.612–2.685)	<0.001
Eating speed	Fast	1.123	<0.001	1.111	<0.001	1.100	<0.001	1.117	<0.001
		(1.105–1.142)		(1.090–1.133)		(1.085–1.116)		(1.100–1.133)	
	Normal	1		1		1		1	
	Slow	0.995	0.597	0.875	<0.001	0.883	<0.001	0.865	<0.001
		(0.986–1.024)		(0.843–0.907)		(0.860–0.907)		(0.841–0.889)	
Skip breakfast	Yes	1.047 (1.024–1.070)	<0.001	1.036 (1.008–1.064)	0.012	1.012 (0.993–1.031)	0.228	0.996 (0.976–1.015)	0.657
Exercise habits	Yes	1.026 (1.007–1.044)	0.006	1.018 (0.996–1.041)	0.105	1.027 (1.010–1.044)	0.001	1.012 (0.995–1.029)	0.164
Physical activity	Yes	0.862 (0.848–0.876)	<0.001	0.859 (0.842–0.876)	<0.001	0.858 (0.846–0.871)	<0.001	0.876 (0.863–0.890)	<0.001
Walking speed	Yes	0.982 (0.968–0.997)	0.017	0.980 (0.963–0.998)	0.03	0.975 (0.962–0.988)	<0.001	0.970 (0.957–0.984)	<0.001
Smoking	Yes	1.414 (1.390–1.438)	<0.001	1.391 (1.362–1.421)	<0.001	1.358 (1.337–1.380)	<0.001	1.351 (1.330–1.374)	<0.001
Frequency of drinking	Daily	1		1		1		1	
	Occasionally	1.241	<0.001	1.204	<0.001	1.232	<0.001	1.187	<0.001
		(1.218–1.265)		(1.177–1.232)		(1.211–1.252)		(1.166–1.207)	
	Almost never	1.583	<0.001	1.526	<0.001	1.568	<0.001	1.496	<0.001
		(1.548–1.619)		(1.484–1.568)		(1.537–1.6)		(1.465–1.528)	
Amount of sake	<180 mL	1		1		1		1	
	180–360 mL	1.016	0.113	1.026	0.038	1.019	0.041	1.020	0.037
		(0.996–1.037)		(1.000–1.502)		(1.000–1.038)		(1.001–1.039)	
	360–540 mL	1.250	<0.001	1.199	<0.001	1.196	<0.001	1.227	<0.001
		(1.218–1.282)		(1.162–1.237)		(1.169–1.224)		(1.198–1.256)	
	>540 mL	1.607	<0.001	1.510	<0.001	1.527	<0.001	1.531	<0.001
		(1.542–1.675)		(1.433–1.591)		(1.471–1.585)		(1.472–1591)	
Rested by sleep	Yes	1.052 (1.036–1.070)	<0.001	1.072 (1.052–1.093)	<0.001	1.060 (1.046–1.075)	<0.001	1.074 (1.059–1.090)	<0.001

CI: confidence interval. Odds Ratio: multivariate logistic regression analysis of region of residence and prevalence of dyslipidemia with obesity, sex, age, meal speed, bedtime meal, nighttime meal, no breakfast, exercise habits, physical activity, walking speed, number of drinks, amount of sake, and rest with sleep as adjustment factors.

## Data Availability

The data used in this study were obtained from the Japan Ministry of Health, Labor, and Welfare through formal procedures. (https://www.mhlw.go.jp/stf/seisakunitsuite/bunya/kenkou_iryou/iryouhoken/reseputo/index.html accessed on 26 July 2022).

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
