# Peer review of "Trends and Regional Differences in the Prevalence of Dyslipidemia before and after the Great East Japan Earthquake: A Population-Based 10-Year Study Using the National Database in Japan"

_ijerph, 2022, doi:10.3390/ijerph20010560_

Round 1

Reviewer 1 Report

Using a national database, this descriptive epidemiological study compares the 10-year trend in the prevalence of dyslipidemia in Fukushima Prefecture prior to the Great East Japan Earthquake with that in evacuated areas and other areas. It is a valuable study describing the long-term health effects associated with a large-scale natural disaster, but there are some problems with the study.

1. Why did the authors focus on dyslipidemia? Although the introduction states that cardiac deaths are high, that measures against atherosclerotic disease are important, and that most papers on deteriorating health status have a short observation period, it is not clear why the focus was on dyslipidemia.

2. What is the National database? When, who, where, and how was the database constructed? And how did the authors obtain the data?

3. The number of analyzed participants is 3859677, but the total of N in Table 1 is only about 2.2 million. Why?

4. Why is LDL cholesterol or total cholesterol excluded from the analysis of dyslipidemia? LDL cholesterol or total cholesterol should have been measured in the specific health checkups.

5. L160-L161 says that triglycerides and HDL were compared using the Krausla-wallis test and medication use and prevalence of dyslipidemia were compared using the χ squared test, but between what were they compared?

6. The text (L232-237) describes comparisons of lifestyle between time phases in each region. However, Table 3 appears to be comparisons of lifestyles among districts by phase.

It would be better to create a table that matches the main text, or to adapt the descriptions in the main text to the table.

7. In Figure 2, the number of municipalities shown in dark red as evacuation areas appears to be 14. However, the main text states that the number of municipalities in the evacuation area is 12. Which is correct?

8. The annual trends for mean TG and mean HDL cholesterol shown in Figure 3 and Figure 4 should be age-adjusted averages.

9. In terms of drinking frequency, the odds ratio for dyslipidemia is higher for Almost never compared to Daily, but in terms of alcohol consumption, the odds ratio for dyslipidemia is higher for alcohol consumption greater than 540 ml. Please explain this discrepancy.

In Table 3, the patients are classified into 2 categories according to the amount of alcohol consumed, and in Table 4, the patients are classified into 4 categories according to the amount of alcohol consumed. Table 5 probably classifies them into two categories, although it is not clear because the category names are not written. Why do the categories differ from table to table?

Also, it is assumed that the analysis of alcohol consumption was performed only for drinkers and non-drinkers were excluded. When multivariate adjustment was made in Table 5, who was analyzed and how were frequency and volume of alcohol consumption treated?

The footnote states that less than 180 ml out of 4 levels was used as a reference, but what are the 4 levels? Is the table correct?

10. Isn't the definition of Obesity BMI ≥25 kg/m2 "or" abdominal circumference?

Reviewer 2 Report

Thank you for asking me to review this article. The topic under study refers to the events following the largest earthquake that hit Japan in 2011, followed by a tsunami and the consequent nuclear crisis in Fukushima. The consequences of this devastating environmental event have had repercussions not only social, economic and in terms of the number of lives lost but also and above all on the health of survivors and displaced persons, also in consideration of the increase in diseases linked to the lifestyle assumed due to changes in the environment.

The theme addressed by the authors is certainly interesting and opens up many food for thought about the evidence that these territories and the populations that populate them are still suffering and fighting for their rebirth with a decisive influence on health and in particular on the incidence of cardiovascular diseases.

Given the topicality of the topic and the interest that the contents arouse, I only think I can provide some small suggestions that the authors might consider considering. In particular, in my opinion, the introductory section was argued in an excessively synthetic way, the authors could consider deepening some of the concepts described and accompanying them with a more complete bibliography to support the hypotheses underlying the question. For example, would it be interesting to know how the displaced populations lived in the period under consideration? In what living conditions? At what distance from the hospital centers? How was health care provided? Were the displaced areas easily accessible? Have health education interventions been carried out? What measures were taken by governments in those years to support displaced populations? This information could be important not only to better clarify the context around which the research question develops but also to open food for thought on possible interventions that can be implemented with a view to future perspectives.

Furthermore, although the investigation ended in the pre-COVID-19 period, under discussion, an aspect not to be overlooked and which should act as a corollary in the background commented on by the authors concerns the pandemic context in progress for which access to services healthcare has undergone significant slowdowns. Infact  for a population plagued by environmental disasters, the difficult of access to health services could have directly and indirectly affected the quality of health care provided, so I would suggest that the authors consider the possibility of mentioning brief comments or considerations about it. 

Author Response

Dear Reviewer

Reviewer 3 Report

Dear authors:

According you stated, your aim was to investigate the effects of post-disaster evacuation life on the prevalence of dyslipidemia in Fukushima Prefecture, which was only partially addressed with a 22-item self-administered questionnaire. Despite this, the work is interesting, the results are well supported by a large amount of data, the weaknesses are adequately addressed by you in the discussion, but the theoretical support of the research is weak. In this regard, the manuscript would greatly benefit of a theoretical frame that explain why dyslipidemia could be affected by evacuation.

Some specific comments are:

- The introduction is weak. Only 1 of the 6 references cited is a reference of a peer-reviewed journal. The reason why you expected that evacuees have differences in the prevalence of dyslipidemia needs a deeper theoretical support. Please add more references.

- In page 3, lines 119-120, add the reference for BMI and waist circumference cut-off points. Also check the reference values for waist circumference, it seems that they are inverted between men and women.

- In Statistical Analysis, have you evaluated the correlation between area of residence and lifestyle-related factors before performing multivariable logistic regression?

- In Results (page 6, lines 203-208) you mentioned the inflection points in trends of prevalence of dyslipidemia, but these are not addressed later in the discussion. What is the importance of these inflection points and how can you explain these inflection points?

- In the Discussion, page 13, lines 342-343, you mentioned that smoking rates have increased since 2011 in women but remained unchanged in men. Please add some reference to support that statement.

-In figure 1, outcome data missing are 6946, not 6947.

- In headings of table 2, what means the number 3 inside the parentheses next to X2?

- Table 3 would be much easier to interpret if you grouped initially by factor and then by phase.

Author Response

Dear Reviewer
